# Identification and Characterization of Salt-Responsive MicroRNAs in *Vicia faba* by High-Throughput Sequencing

**DOI:** 10.3390/genes10040303

**Published:** 2019-04-17

**Authors:** Saud M. Alzahrani, Ibrahim A. Alaraidh, Muhammad A. Khan, Hussein M. Migdadi, Salem S. Alghamdi, Abdluaziz A. Alsahli

**Affiliations:** 1Botany and Microbiology Department, College of Science, King Saud University, Riyadh 11451, Saudi Arabia; saood122@hotmail.com (S.M.A.); ialaraidh@ksu.edu.sa (I.A.A.); aalshenaifi@ksu.edu.sa (A.A.A.); 2Plant Production Department, College of Food and Agriculture Sciences, King Saud University, Riyadh 11451, Saudi Arabia; h.migdadi@gmail.com (H.M.M.); Salem@ksu.edu.sa (S.S.A.); 3Plant Biotechnology Department, National Agricultural Research Center, Baq’a 19381, Jordan

**Keywords:** microRNAs, Hassawi-3, ILB4347, faba bean, salt stress

## Abstract

Salt stress has detrimental effects on plant growth and development. MicroRNAs (miRNAs) are a class of noncoding RNAs that are involved in post-transcriptional gene expression regulation. In this study, small RNA sequencing was employed to identify the salt stress-responsive miRNAs of the salt-sensitive Hassawi-3 and the salt-tolerant ILB4347 genotypes of faba bean, growing under salt stress. A total of 527 miRNAs in Hassawi-3 plants, and 693 miRNAs in ILB4347 plants, were found to be differentially expressed. Additionally, 284 upregulated and 243 downregulated miRNAs in Hassawi-3, and 298 upregulated and 395 downregulated miRNAs in ILB4347 plants growing in control and stress conditions were recorded. Target prediction and annotation revealed that these miRNAs regulate specific salt-responsive genes, which primarily included genes encoding transcription factors and laccases, superoxide dismutase, plantacyanin, and F-box proteins. The salt-responsive miRNAs and their targets were functionally enriched by Gene Ontology (GO) and Kyoto Encyclopedia of Genes and Genomes (KEGG) pathway analyses, which showed that the miRNAs were involved in salt stress-related biological pathways, including the ABC transporter pathway, MAPK signaling pathway, plant hormone signal transduction, and the phosphatidylinositol signaling system, among others, suggesting that the miRNAs play an important role in the salt stress tolerance of the ILB4347 genotype. These results offer a novel understanding of the regulatory role of miRNAs in the salt response of the salt-tolerant ILB4347 and the salt-sensitive Hassawi-3 faba bean genotypes.

## 1. Introduction

Salt stress is one of the major abiotic stresses that negatively threaten plant growth and development. Worldwide, about 20% of agricultural lands and 50% of croplands are exposed to salt stress [1]. High salinity also results in secondary stresses, such as nutritional imbalance and oxidative stress, that lead to cellular damage, inhibition of growth, and a reduction in crop yield. *Vicia faba* is an economically important pulse. Owing to its high nutritional value, it is commonly used as food material and is widely distributed throughout the world. Understanding the role of *V. faba* microRNAs (miRNAs) in response to salt stress may help screen gene function and regulation in leguminous plants, and could contribute to more effective plant breeding.

The miRNAs comprise a class of endogenous non-coding RNAs that are approximately 21–24 nucleotides (nts) long, and are recognized as an important class of regulatory molecules, involved in post-transcriptional gene regulation mediated via mRNA degradation or the repression of mRNA translation [2,3]. These RNAs are largely derived from intergenic regions and are produced from single-stranded primary miRNAs with unique hairpin structures [4]. These were originally discovered in *Caenorhabditis elegans* [5]. However, their widespread presence has been reported in animals [6], plants [7], and certain viruses [8]. It is generally known that miRNAs play important regulatory roles in several biological processes, including those during the developmental, metabolism, pathogen defense, and stress responses in plants [9,10]. Recently, it was discovered that miRNAs respond to various abiotic stresses in plants, including drought [11,12,13,14], high salinity [15,16,17], low temperatures [16,18], oxidative stress [19], hypoxic stress [20,21], mechanical stress [22], and UV-B radiation [15], as well biotic stresses [23].

Multiple gene expression mechanisms have evolved in plants in order to address high-salinity-induced stress. These mechanisms relate to a broad spectrum of biochemical, cellular, and physiological processes, including signal transduction, energy metabolism, transcription, protein biosynthesis, membrane trafficking, and photosynthesis [24]. The transcriptional regulation of several miRNAs and genes in response to high salt stress has been extensively studied [25,26]. These studies suggested that plant responses to salt stress could be shaped by miRNA-guided gene regulation. Microarray analysis revealed that several miRNAs, such as miR156, miR159, miR167, miR168, miR171, miR319, and miR396, showed differential expression during the salt stress response of *Arabidopsis* sp. [16] and *Zea mays* [27]. Wide-ranging sequencing data has been produced from next generation sequencing (NGS) technology for the detection of salt-responsive miRNAs in various plant species. Employing this technology, 104 differentially expressed miRNAs were identified in the salt-stressed functional soybean nodules [28].

In this study, we employed high-throughput sequencing technology to identify the conserved and novel miRNAs in the salt-tolerant and salt-sensitive cultivars of *V. faba* growing under conditions of high salt stress. Using advanced bioinformatics analysis, the changes in miRNA expression following salt treatment were studied in comparison to those of the control. To study the potential underlying mechanism of the miRNA-mediated gene expression regulation in faba bean under salt stress, the miRNA and target interaction networks were further analyzed by Gene Ontology (GO) enrichment and Kyoto Encyclopedia of Genes and Genomes (KEGG).

## 2. Materials and Methods

### 2.1. Plant Materials, Growth Conditions, and Salt Stress Treatments

Two genotypes of *V. faba*, namely the salt-sensitive Hassawi3 and the salt-tolerant ILB4347, growing under control and stress conditions were used in this study. The plants from both the genotypes were washed in sterile deionized water and left to germinate in sand at 23 °C for 5 days. The seedlings that had germinated homogeneously were selected and planted in a mixture of sterilized sand and peat moss at a ratio of 3:1. The experiment was conducted in a greenhouse with a 16 h light/8 h dark cycle at a 26 °C/20 °C day/night temperature and 50–80% relative humidity at the College of Science, King Saud University, Riyadh, Saudi Arabia. The seedlings were irrigated with tap water and allowed to grow for 14 days. Hoagland nutrient solution (1/10 strength) was applied once a week. NaCl stress was initiated when the seedlings were nearly 21 days old and had attained two to three true leaves. Treatments constituted the control group and the groups that were administered NaCl at 150 mM concentrations. For both genotypes, the samples for RNA extraction were collected after two weeks of treatment, when the stressed plant started to wilt in comparison to the plants in the control setup and the plants that received 150 mM NaCl and had been exposed to maximum stress. The collected samples were quickly frozen in liquid nitrogen and stored at −80 °C until RNA extraction. A pool of three replicates was used for small RNA (sRNA) sequencing. HC-4 and HSA represented the Hassawi3 genotype growing under control and stress conditions, respectively. Similarly, IC-1 and IS-4 represented the ILB4347 genotype growing under control and stress conditions, respectively.

### 2.2. sRNA Library Construction and Sequencing

The total RNA was extracted from the foliar tissues of the salt-sensitive Hassawi-3 and the salt-tolerant ILB4347 genotypes of faba bean using the total RNA extraction kit (SV Total RNA isolation system, Promega, Madison, WI, USA), according to the manufacturer’s instructions that involved the on-column digestion of any residual DNA in the samples of the control and stress-induced plants with RNase-free DNase I. The quality and quantity of the extracted RNA were measured using a NanoDrop ND-2000 spectrophotometer (Thermo Scientific, Wilmington, DE, USA). Then, the RNAs were sent to BGI (Shenzen, China) for sRNA library construction and high-throughput sequencing using BGISEQ-500 technology [29,30].

### 2.3. Data Filtering and Mapping Reads

The raw reads were first cleaned; this involved the removal of low-quality tags, tags without a 3′ primer, tags with 5′ primer contaminants, tags without the insert, tags with poly A, and tags that were shorter than 18 nts. The length distribution of the cleaned reads was next categorized for analyzing the composition of the sRNA data, and the 16–28-nt long sRNAs were selected for further analyses. The tags that remained after filtering comprised the “clean tags”, which were stored in FASTQ format.

Then, the clean, high-quality sRNA tags were mapped to the reference genome using AASRA [31] and other sRNA databases, except Rfam, by using CMsearch [32].

### 2.4. Classification of sRNAs

Some sRNA tags can map to more than one category after alignment and annotation. In order to ensure that each unique sRNA maps to only one category after annotation, we followed the following priority rule: miRNA > piRNA > snoRNA > Rfam > other sRNAs.

### 2.5. Predictions of sRNAs

The typical hairpin structure of the miRNA precursor can be used to forecast novel miRNAs. We used RIPmiR [33] to predict novel miRNAs by analyzing their secondary structures, the Dicer cleavage site, and the minimum free energy of the unannotated sRNA tags, which could be mapped to the genome.

The piRNAs were predicted by Piano [34], which is based on an algorithm that uses piRNA-transposon interaction information, and the support vector machine (SVM) on these features. Small interfering RNAs (siRNAs) [35] are 22–24-nt-long double-stranded RNAs, in which each strand is 2-nt longer than the other at the 3′ end. We aligned the tags on the basis of this structural feature to identify the siRNAs that met this criteria, as these tags could be potential siRNA candidates.

### 2.6. Analyzing sRNA Expression

The sRNA expression level of each gene was estimated by the Transcripts per Kilobase Million (TPM) method [36]. The TPM method can eliminate the influence of sequencing discrepancy while calculating sRNA expression. The gene expression thus calculated can therefore be directly used to compare the differences in gene expression among different samples. The following formula is used to calculate TPM:$$TPM=\frac{C\times 10^{6}}{N}$$

### 2.7. Target Prediction

The computational prediction of miRNA targets is a critical step in the initial identification of miRNA:mRNA target interactions for experimental validation. In this study, psRobot [37] and TargetFinder [38] were used to identify the possible targets.

### 2.8. Screening the Differentially Expressed sRNAs (DESs)

With reference to “the significance of digital gene expression profiles” [39], a stringent algorithm was developed for identifying the differentially expressed genes between the two samples. If the number of tags from an sRNA is denoted as “x”, and given the fact that the expression product of each RNA occupies only a small portion of the library, then the Poisson distribution can be computed from “x” by the following equation:$$P\left(x\right)=\frac{e^{-\lambda }\lambda^{x}}{x!}$$

If the total number of clean tags from sample 1 and sample 2 is N1 and N2, respectively, and if an sRNA “A” contains “x” tags in sample 1 and “y” tags in sample 2, then the probability of sRNA A being equally expressed between the two samples can be calculated as follows:$$2\overset{i=y}{\underset{i+0}{\sum }}P(i|x)$$
or
$$2\;\times \;(1-\overset{i=y}{\underset{i+0}{\sum }}P(i|x)\;(if\;\overset{i=y}{\underset{i+0}{\sum }}P(i|x)>0.5)$$
$$P(y\;|x)={\left(\frac{N_{2}}{N_{1}}\right)}^{y}\frac{\left(x+y\right)!}{x!y!{\left(1+\frac{N_{2}}{N_{1}}\right)}^{x\;\left|\;y\;\right|1}}$$

We corrected the *p*-value corresponding to differential gene expression tests using the Bonferroni method [40]. As DESs analysis generates large multiplicity problems in which thousands of hypotheses are simultaneously tested, such as whether a gene “x” is differentially expressed between two groups, false positive (type I) and false negative (type II) errors are corrected by the false discovery rate (FDR) method [41]. Assuming that among “R” number of differentially expressed genes, “S” number of genes really show differential expression, and the remaining “V” genes are false positive, and supposing that the error ratio Q = V/R must remain below a certain cutoff, say 5%, then the value of FDR should be preset to a number no larger than 0.05. In this study, we preset the value of FDR to ≤0.001, and the absolute value of Log2Ratio was ≥1, which were set as default thresholds for judging the significance of the differences in gene expression. More stringent criteria with smaller values of FDR and higher fold-change values can be used to identify DESs.

### 2.9. GO Enrichment Analysis

Gene ontology (GO) enrichment analysis provides all the GO terms that are significantly enriched in a list of genes, and filters the genes with specific biological functions. In this method, all the genes are first mapped to the GO terms in the database (http://www.geneontology.org/), and then the gene numbers for each term are calculated. Lastly, a hypergeometric test is performed to identify the significantly enriched GO terms in the input gene list, based on GO: TermFinder’ (https://www.yeastgenome.org/goTermFinder). The aforementioned algorithm was used for this analysis, and the P value was calculated by the following equation:$$Ρ=1-\overset{m-1}{\underset{i=0}{\sum }}\frac{\left(\begin{array}{c} M\\ i \end{array}\right)\left(\begin{array}{c} N-M\\ n-i \end{array}\right)}{\left(\begin{array}{c} N\\ n \end{array}\right)}$$
where, N is the total number of genes with GO annotation, n is the number of DESs target genes in N, M is the total number of genes that are annotated by certain GO terms, and m is the number of DESs target genes in M. The calculated *p*-value was corrected by the Bonferroni Correction method [40], considering the corrected *p*-value ≤0.05 as a threshold. The GO terms that fulfill this condition are commonly defined as the significantly enriched GO terms. This analysis is performed to identify the main biological functions of the targets of the DESs.

### 2.10. Pathway Enrichment Analysis

Pathway-based analysis is helpful in further understanding the biological functions of the target genes of the DESs. KEGG [42] is an important public pathway-related database used for performing pathway enrichment analysis. Using this analysis, the significantly enriched metabolic pathways and signal transduction pathways of the DESs target genes are identified by comparing them with the whole genome as the background. The formulae used in pathway-based analysis are the same as those used in GO analysis, where “N” represents the total number of genes with KEGG annotation, “n” represents the total number of DESs target genes in N, “M” represents the total number of genes annotated by specific pathways, and ‘m’ depicts the number of DESs target genes in M.

## 3. Results

### 3.1. sRNA Sequencing

In order to study the salt stress-responsive miRNAs in the salt-sensitive Hassawi-3 and salt-tolerant ILB4347 genotypes of faba bean, four sRNA libraries were constructed from the leaves. These libraries were then sequenced using BGISEQ-500 technology [29,30]. Table 1 briefly summarizes the sequencing information data derived from each sample. Upon sequencing the four libraries, a total of 197147801 raw reads were obtained; with the number of reads from the HC-4, HSA, IC-1, and IS-4 libraries being 51420111, 43734868, 48229628, and 53763194, respectively. The resulting raw sequence reads were computationally processed to remove the 3′ adapter, which yielded a total of 174053893 clean reads from the four libraries; with the number of reads from the HC-4, HSA, IC-1, and IS-4 libraries being 47942551, 40682196, 43865674, and 41563472, respectively. The clean tags were used for analyzing the distribution of the 16–30 nt-long sRNAs. Generally, the length of sRNAs varies between 16–30 nt. Figure 1 shows the results of the length distribution analysis for the samples of sRNA. The length of sRNAs for HC-4 ranged between 17–34 nt, with the 19–30 nt-long RNAs being the most abundant. Similarly, for HSA, IC-1, and IS-4, the lengths of the sRNAs were 17–33 nt, 17–29 nt, and 17–31 nt, respectively. The sRNA length distribution (16–30 nt) of each library indicated that the species with a 21–24 nt length were the most abundant and diverse. The 24-nt-long RNAs were the most diverse among all the four sequencing libraries, followed by the 21-nt-long RNAs that were the second most abundant group of RNAs among the four libraries, with the exception of the HC-4 library, where the 21-nt-long RNAs were the most abundant.

After filtering, the clean tags were mapped to different sRNA databases, such as miRBase [43], Rfam [44], siRNA, piRNA, and snoRNA, among others. Table 1 enlists the individual mapping rates for each sample, while the distribution of the tags is depicted in Figure 1.

The sRNA reads were subsequently annotated by the RFam database (http://rfam.sanger.ac.uk/) and miRbase v21.0 (http://www.mirbase.org/) for classification. Table 2 shows the proportions of different types of sRNAs. The sRNAs were classified into different annotation categories of non-coding RNAs, including miRNA, tRNA, rRNA, snRNA, and snoRNA, among others. In order to ensure that each unique sRNA mapped to only one annotation category, the following priority rule was used: miRNA > piRNA > snoRNA > Rfam > other sRNA.

### 3.2. Annotation of sRNAs and miRNA Identification

A total of 1062 and 1156 miRNAs were identified from HC-4 and HSA, respectively; from which, 44 and 38 novel miRNAs were identified from HC-1 and HSA, respectively (Table 3). A total of 1513 and 1352 miRNAs were identified from IC-1 and IS-4, respectively; of which, 48 and 51 were novel miRNAs, identified from IC-1 and IS-4, respectively (Table 3). Interestingly, none of the siRNAs identified from the four samples were known siRNAs, with all of them being novel in nature.

### 3.3. Identification of Differentially Expressed miRNAs

DESs screening is primarily used to identify the sRNAs that are differentially expressed between different samples, followed by subsequent analysis. The DESs in each pairwise alignment are shown in Figure 2. Among the differentially expressed miRNAs, 284 were upregulated, while 243 miRNAs were downregulated in HC-4 and HSA combination (Figure 2). Similarly, 298 of the differentially expressed miRNAs were upregulated in IC-1 and IS-4 combination, while 395 were found to be downregulated (Figure 2).

The miRBase database v21.0 was used to identify the conserved miRNAs in our study from the perfect or near-perfect matches, with the latter having up to two mismatches. Based on sequence similarity, our analysis identified 492 known differentially expressed miRNAs in the salt-sensitive Hassawi-3 genotype (HC-4-vs.-HSA), belonging to 39 miRNA families (Appendix A). Of these 39 miRNA families, seven families, namely those of miR166, miR167, miRNA396, miRNA159, miRNA171, miRNA168, and miR156, contained 130, 64, 52, 35, 33, 32, and 26, miRNAs, respectively; while 16 families contained two to 12 miRNAs, and 16 other families were represented by a single miRNA (Appendix A). The majority of miRNAs in 25 of these families were found to have been upregulated (Table 4). A total of 35 sRNAs were identified as putative novel, faba bean miRNAs (Appendix A and Table 5), of which 22 were found to be upregulated and 13 were downregulated (Table 5).

Similarly, in the salt-tolerant genotype ILB4347 (IC-1-vs.-IS-4), 665 known miRNAs that belonged to 31 miRNA families were found to have been differentially expressed (Appendix A). Of these 31 miRNA families, nine families, of miR166, miR167, miRNA396, miRNA160, miRNA171, miRNA164, miRNA159, miRNA168, and miR408, contained 162, 58, 57, 54, 54, 45, 36, 33, and 21 miRNAs, respectively; while 12 families contained two to twelve miRNAs, and 10 families were represented by a single miRNA (Appendix A). The majority of miRNAs in 21 of these families were found to have been downregulated (Appendix A). Additionally, 28 sRNAs were found to be novel miRNAs (Table 6), of which 11 were found to have been upregulated and 17 were downregulated (Table 6).

The distribution miRNAs in these two genotypes showed that some of the miRNAs present in other legumes are also present in vicia faba or vice versa. The miRNAs such as 1507, 1508, 1509, 1512, 1514, 1521, 2086, 2109, 2199, miR2111, R2118, R5213 and R5232, 4414, 5213,5232, and 5234,5770 were generally found in legumes such as *Cicer arietinum, Vigna unguiculata, Glycine max, Medicago truncatula*, *Glycine soja*, *Lotus japonicus*, *Phaseolus vulgaris, Lotus japonicus*, and *Acacia auriculiformis* (www.miRBase.com). Of these miRNAs, 5213, 5232, 5234, and 21111, were found in either of our two faba bean genotypes (Appendix A). However, the miR2111 was not only reported in legumes, but was also found in other plants such as in *Populus trichocarpa, Vitis vinifera, Arabidopsis thaliana,* and *Malus domestica* (www.miRBase.com). The reason for the presence of a limited number of legume-specific miRNAs in our study is that our small RNA library is not comprehensive owing to one tissue source under abiotic stress. Therefore, miRNAs in other tissues under biotic and abiotic stresses could speculate whether other miRNAs frequently present in legumes are also expressed in faba bean.

### 3.4. Target Prediction and Functional Analysis of the miRNAs

In the HC-4-vs.-HSA combination, a total of 4996 putative targets were predicted for 527 miRNAs; 4972 targets were of known miRNAs and 72 were of novel miRNAs (Appendix A). For the IC-1 vs. IS-4 combination, a total of 5785 putative targets were predicted for 693 miRNAs; 4910 targets were of known miRNAs and 62 were of novel miRNAs (Appendix A). Most of the predicted target genes encoded some stress-related transcription factors (TFs), including those of the auxin response factor (ARF) family, the AP2-like ethylene-responsive TF, squamosa promoter-binding proteins, myb domain proteins (MYBs), NAC domain-containing proteins, nuclear transcription factor Y, and bZIP proteins (Appendix A), as well as other proteins such as Argonaute (AGO2), glutamate decarboxylase, laccases, superoxide dismutase, plantacyanin, and F-box proteins.

For understanding the biological functions of miRNAs, all the putative target genes were subjected to GO functional classification by the Blast2GO software. GO-based enrichment analysis was carried out to further probe the possible role of miRNAs in response to salt stress. In the HC-4 vs. HSA combination, a total of 3708 potential miRNA targets were mapped to 1651 biological processes, 661 molecular functions, and 1396 cellular components (Figure 3). Some of the significant biological processes with the highest target numbers were the cellular process (348), metabolic process (318), biological regulation (148), regulation of the biological process (134), response to stimulus (102), and signaling (43). Among the categories for molecular function, binding (297), catalytic activity (270), transporter activity (23), nucleic acid binding TF activity (18), molecular transducer activity (8), and protein binding TF activity (6) were the most abundant classes (Figure 3). The common cellular component terms were cell (305), cell part (304), organelle (235), and membrane (201).

In the IC-1-vs.-IS-4 combination, a total of 3354 potential miRNA targets were mapped to 1488 biological processes, 574 molecular functions, and 1292 cellular components (Figure 4). Some of the significant biological processes with the highest target numbers were the cellular process (326), metabolic process (291), biological regulation (121), regulation of the biological process (102), response to stimulus (93), and signaling (34). Among the categories for molecular function, binding (253), catalytic activity (246), transporter activity (21), nucleic acid binding transcription factor activity (17), molecular transducer activity (4), and structural molecule activity (13) were the most abundant classes (Figure 4). The common cellular component terms were cell (278), cell part (278), organelle (232), and membrane (177).

For the HC vs. HSA combination, pathway analysis was performed by KEGG annotation, which returned 93 KEGG pathways that were classified into five main categories (Figure 5). The complete set of 93 pathways for the HC vs. HSA combination after KEGG annotation has been summarized in Appendix A. The annotation results showed that most of the genes were involved in “metabolic function” (327 unigenes), followed by those involved in “genetic information processing” (153), “environmental information processing” (67), “cellular processes” (52), and “organismal systems” (12). The only four pathways that were categorized under the “environmental information processing” category were from plants (24, 3.69%). The majority of representative pathways for the unigenes under the “metabolism” category were involved in the biosynthesis of secondary metabolites (ko01110; 45, 6.92%). The two pathways categorized under the “organismal systems” (environmental adaptation) category were involved in plant-pathogen interactions (ko04626; 10, 1.54%) and plant circadian rhythms (ko04712; 2, 0.31%).

For pathway analysis, KEGG annotation was performed for the IC-1 vs. IS-4 combination, which returned 89 KEGG pathways that were classified into five main categories (Figure 6). The complete set of 89 pathways identified for the IC-1 vs. IS-4 combination is summarized in Appendix A. The majority of genes were annotated under the “metabolic function” (243) category, followed by the “genetic information processing” (123) category, “environmental information processing” (54), “cellular processes” (33), and “organismal systems” (9) categories. The only four pathways under the “environmental information processing” category were categorized under the plant hormone signal transduction (ko04075; 34, 6.53%), plant MAPK signaling pathway (ko04016; 6, 1.15%), ABC transporter (ko02010; 3, 0.58%), and phosphatidylinositol signaling system (ko04070; 13, 2.5%) categories. The majority of representative pathways for the unigenes under the “metabolism category” were involved in metabolic pathways (ko01100; 76, 14.59%), and the biosynthesis of secondary metabolites (ko01110; 30, 5.76%). The two pathways categorized under the “organismal systems” (environmental adaptation) category were involved in plant-pathogen interactions (ko04626; 9, 1.73%).

## 4. Discussion

Salinity adversely affects plant growth and development. Therefore, in order to cope with salt stress, plants adapt themselves by modulating various stress-responsive genes at the transcriptional and post-transcriptional levels. In recent years, the role of miRNAs in the regulation of gene expression has been increasingly understood. The miRNAs are said to have pivotal roles in plant responses to abiotic and biotic stresses [45]. Numerous studies have demonstrated that miRNA-mediated gene regulation plays an important role in the salt stress response of several plant species, including Arabidopsis [16], soybean [11], barley [46], and sugarcane [47].

In this study, we investigated millions of sRNA sequences from faba bean for understanding the effects of salt stress on plant growth and development under miRNA regulation. Our results revealed that salt stress modulated the expression of miRNAs and their predicted targets in faba bean.

More than 40 million sRNA reads were produced by high-throughput sequencing from each library of faba bean genotype growing under control and salt stress conditions. The sRNAs of faba bean comprised different classes of RNAs, including snRNAs, tRNAs, snoRNAs, rRNAs, and miRNAs. The majority of these were unannotated in the existing sRNA libraries owing to the scarcity of information on this species. The annotation performed herein was in accordance with numerous previous studies [48,49]. The results indicated that to date, numerous sRNAs remain to be identified. Among the different types of sRNAs, miRNAs are considered as being crucial in the regulation of gene expression at the post-transcriptional level. In plants, the genes under miRNA regulation are involved in vegetative and reproductive growth, as well as in the plant stress response [49,50]. Analysis of the four sRNA libraries of faba bean revealed that most of the sRNAs were 21-nt and 24-nt-long (Figure 1). This pattern of length distribution has been observed in several other plant species, including mulberry [49], *Populus euphratica* [51], potato [52], and barley [53]. The 24-nt-long miRNA class was the most diverse among all the four libraries, except for HC-4, where the 21-nt-long miRNAs were the most abundant. Other studies have indicated that the length of miRNAs primarily varies between 21 nt and 22 nt, while siRNAs are mostly 24-nt-long [4,54]. Our study therefore revealed that the more enriched faba bean sRNAs could in reality be miRNAs that had been cleaved by DCL1. The importance of the miRNA length distribution lies in the fact that it allows easy alignment with the RNA-induced silencing complex (RISC) for regulating gene expression by inhibiting its translation or by degrading the target mRNAs, depending on the miRNA-mRNA similarities.

### Salt Stress-Responsive miRNAs and Their Targets in Faba Bean

Recently, miRNAs have evolved as a valuable genetic tool for interpreting stress tolerance at the molecular level and for finally regulating the stress response in crops [55]. The identification of salt stress-responsive miRNAs and their subsequent functional analysis will aid the understanding of the stress-responsive mechanism in plants. Earlier studies in maize [27], wheat [56], rice [57], barley [46], and sugarcane [47] have demonstrated that miRNAs such as miR156, miR169, miR160, miR159, miR168, miR171, miR172, miR393, and miR396 are the major salt stress-responsive miRNAs in plants. In our study, we identified 527 differentially expressed miRNAs in the HC-4 vs. HSA combination; 284 of which were upregulated and 243 were downregulated (Appendix A). A total of 693 differentially expressed miRNAs were identified in the IC-1 vs. IS-4 combination; 298 of which were upregulated and 395 were downregulated (Appendix A). The aforementioned salt stress-responsive miRNAs were also identified in our study, which suggests the presence of common salt stress-related miRNAs. Additionally, other conserved miRNAs and a few novel miRNAs were identified as salt stress-responsive sRNAs in both genotypes (Table 5 and Table 6). Therefore, these results serve as novel information on the salt stress-responsive miRNAs of plants. It is well-established that miRNA-mediated complex regulatory networks are involved in the regulation of gene expression at the post-transcriptional level in plants [58]. The GO terms of the putative targets revealed that the targets had stimulus signaling, binding, catalytic, transporter, and nucleic acid binding TF activities (Figure 3 and Figure 4). KEGG analysis revealed that some targets of the salt stress-responsive miRNAs in the two faba bean genotypes considered in this study were mapped to salt stress-related pathways, such as plant hormone signal transduction, flavonoid biosynthesis, ABC transporter activity, ubiquitin-mediated proteolysis, and DNA repair (Figure 5 and Figure 6, Appendix A) [59].

The large number of upregulated and downregulated miRNAs induce a more pronounced change in the expression of multiple downstream genes. The response to salt stress in crops is accompanied by a wide range of intracellular processes, including signal perception, signal transduction, transcription, and protein biosynthesis, among others. Although different species of plants respond to stress by employing numerous miRNA-mediated regulatory strategies [60], some studies have reported that hub miRNAs, such as miR169, miR171, miR393, miR396, and miR398, among others, are linked to several abiotic stresses, such as drought, salinity [61,62], and cold [63]. These studies report that the miRNA targets are involved in sensing stress, signal transduction, and other stimuli. Previous studies have demonstrated that miR171 and miR393 are upregulated under conditions of salt stress in *A.* sp., wheat, and barley [46,64,65]. In this study, we observed that the expression of miRNAs in both the miR393 and miR171 families was upregulated and downregulated under conditions of salt stress in all the genotypes of faba bean, except for the miR393 family, in which the miRNA expression was only upregulated for the HC-4 vs. HSA combination. These results suggest the existence of common regulatory mechanisms for salt stress response in plants, and these miRNAs might control the same targets in different plants [66]. The miR393 family targets genes encoding the F-box protein family that includes TIR1 and AFB2 in *A.* sp. and rice, and inhibits the growth of lateral roots under abscisic acid (ABA) treatment or osmotic stress [65,67].

In this study, miR166, miR171, miR398, miR396, and miR1432 were identified in both genotypes of faba bean. The miR171 family is known to target the myeloblastosis (MYB) family of TFs that might have a role in the regulation of osmotic balance under drought and salt stress conditions. Alptekin and coworkers (2017) reported that both salt stress and drought affect the osmotic balance of plant cells [62]. In this study, miR396 was found to be differentially expressed in both the genotypes of faba bean. Using expressed sequence tags (ESTs), Kantar and coworkers (2011) reported that the miR396 family targets the growth factor-like (GRL) TF and its putative heat-shock protein, as the changes in their expression were consistent with the changes in the expression of miR396. The study demonstrated that the expression of GRL TF and its putative heat-shock protein was upregulated following the downregulation of miR396 under stress [68]. The function of heat-shock proteins is to protect other proteins from degradation under stress. The downregulation of miR396 and the subsequent regulation of its targets would therefore improve the tolerance of faba bean to salt stress. A better understanding of miRNAs and their functions under stress conditions would improve the efficacy and reliability of the application of miRNA-mediated gene regulation for increasing plant stress tolerance.

## 5. Conclusions

To the best of our knowledge, this is the first study to identify the salt stress-responsive miRNAs in *Vicia faba* using high-throughput sequencing technology. Four sRNA libraries were generated and sequenced from two faba bean genotypes under control and salt stress conditions. A total of 1062 and 1156 miRNAs were identified in the HC-4 and HSA samples, respectively; while a total of 1513 and 1352 miRNAs were identified in IC-1 and IS-4, respectively. Differential expression analysis revealed that 527 miRNAs were differentially expressed in the HC-4 vs HSA combination; whereas 693 miRNAs were differentially expressed in the IC-1 vs. IS-4 combination. GO enrichment analysis revealed that the targets of these differentially expressed salt stress-responsive miRNAs were highly enriched in the corresponding GO terms, and possessed stimulus signaling, binding, catalytic, transporter, and nucleic acid binding TF activities, among others. KEGG analysis suggested that several of the miRNAs in faba bean participate in stress-related pathways, including plant hormone signal transduction, the MAPK signaling pathway, flavonoid biosynthesis, ubiquitin-mediated proteolysis, apoptosis, and ABC transporter activity. This study enhanced the existing genetic information and resources of salt-responsive miRNAs in faba bean. This will not only improve our understanding of the roles of miRNAs in post-transcriptional gene regulation during salt stress response, but will also aid studies on the miRNA-based genetic improvement of salt tolerance in cultivated faba bean genotypes.

## Figures and Tables

**Figure 1 genes-10-00303-f001:**
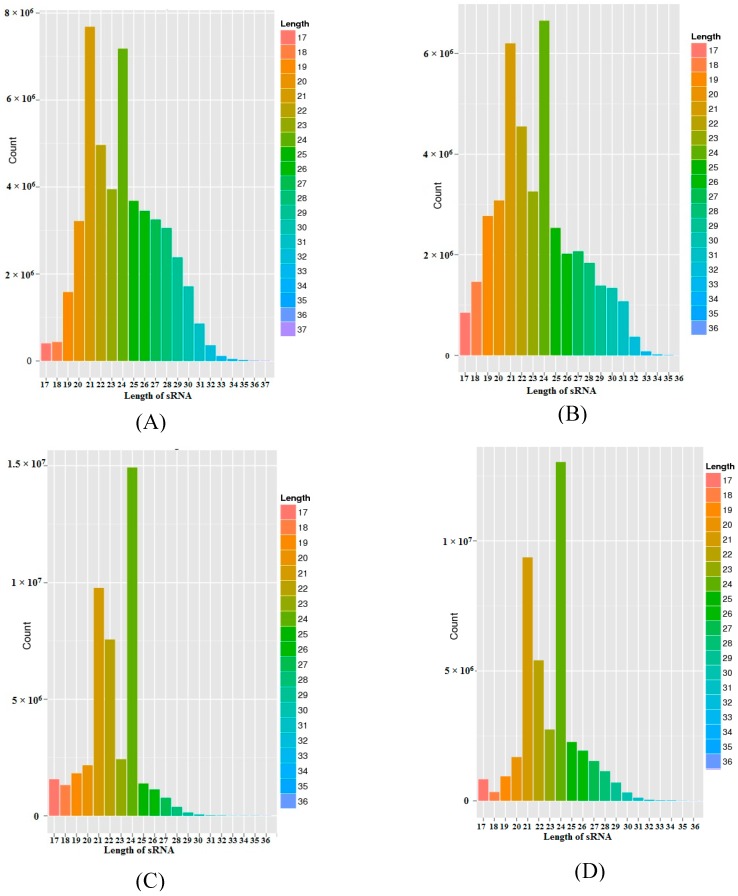
Sequence length distribution of the faba bean sRNAs. The size distribution of the sRNA libraries. The y-axis represents the sequences, while the x-axis depicts the 16–36 nt-long sequences for each of the four sequenced libraries. Length distribution of (**A**) HC-4, (**B**) HSA, (**C**) IC-1 and (**D**) IS-4 libraries.

**Figure 2 genes-10-00303-f002:**
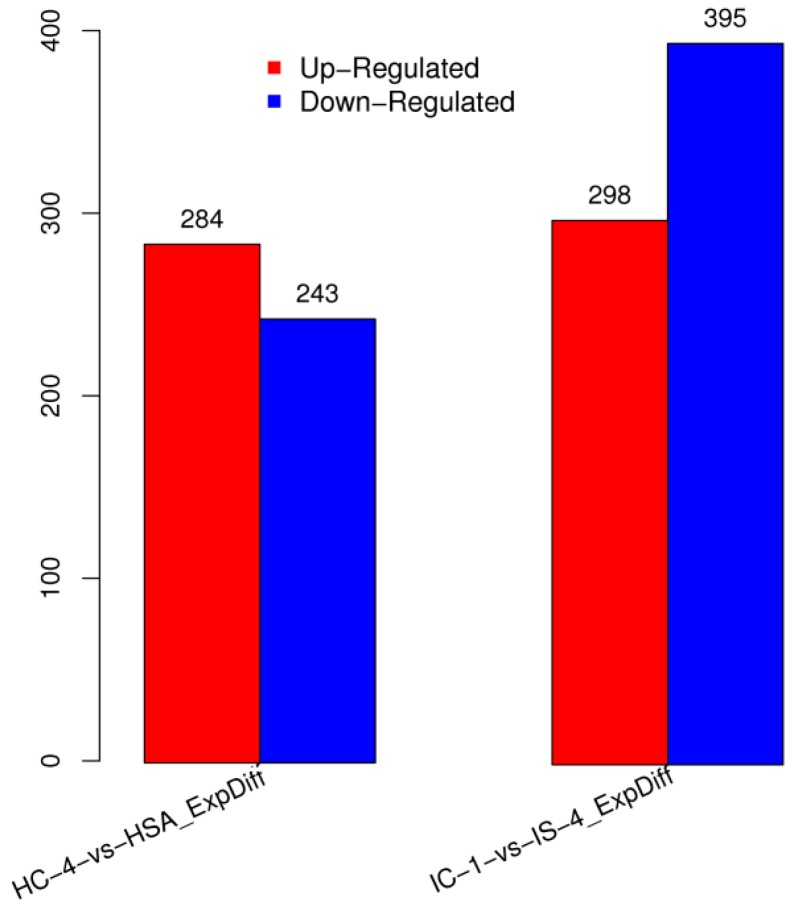
Graphical representation of the number of differentially expressed miRNAs in salt-sensitive (HC-4 vs HSA) and salt-tolerant genotypes (IC-1 vs IS-4) growing under control and salt stress conditions.

**Figure 3 genes-10-00303-f003:**
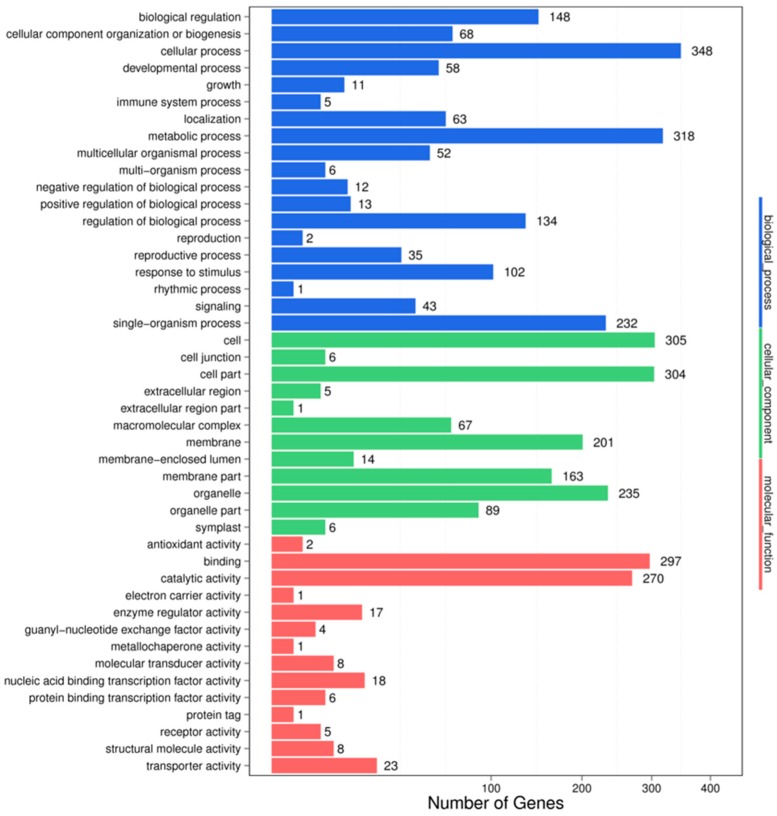
GO classifications in the HC-4 vs. HSA combination.

**Figure 4 genes-10-00303-f004:**
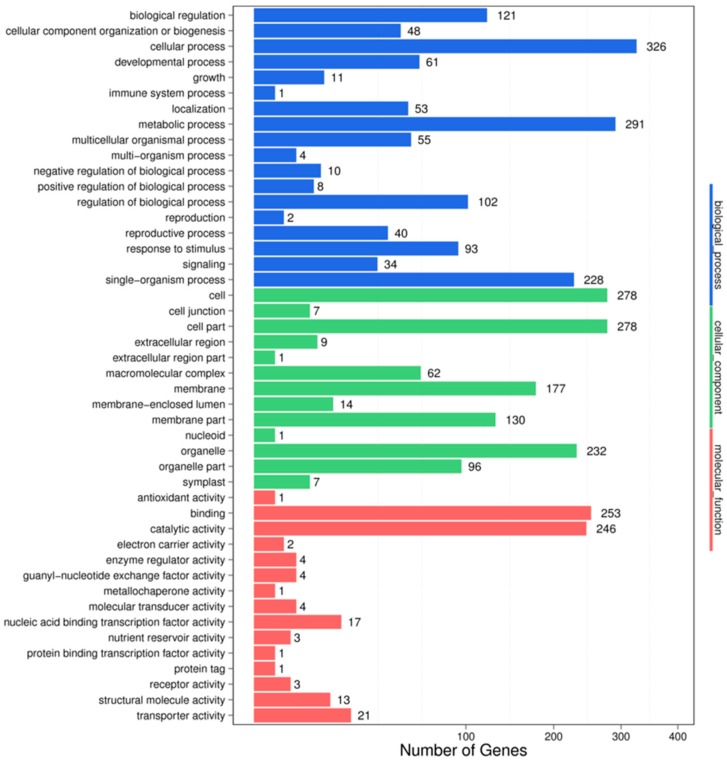
GO classifications in the IC-1 vs. IS-4 combination.

**Figure 5 genes-10-00303-f005:**
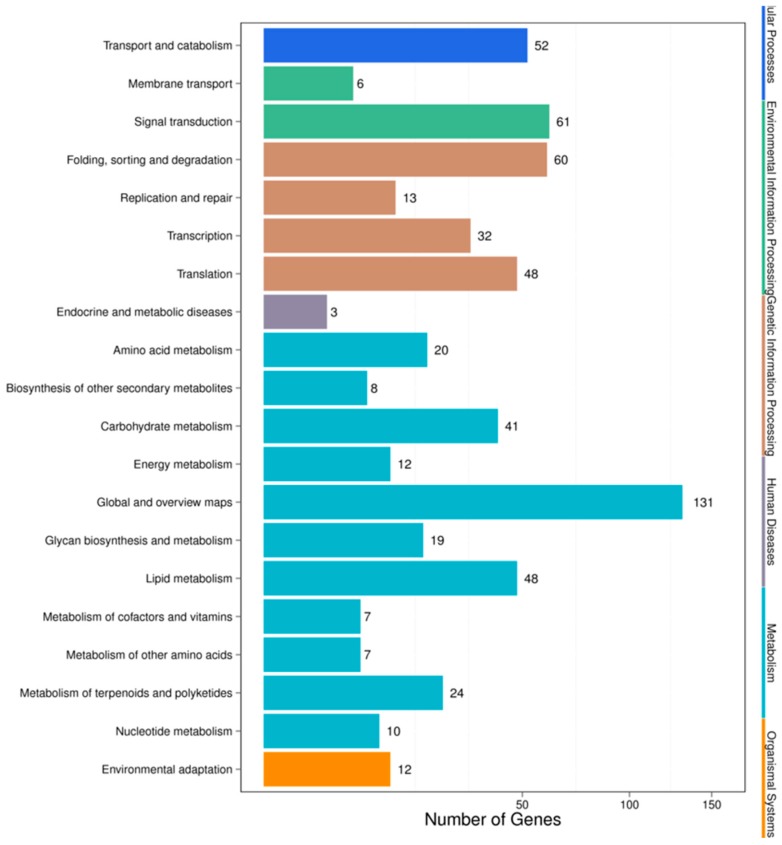
KEGG pathway classifications of the HC vs. HSA combination.

**Figure 6 genes-10-00303-f006:**
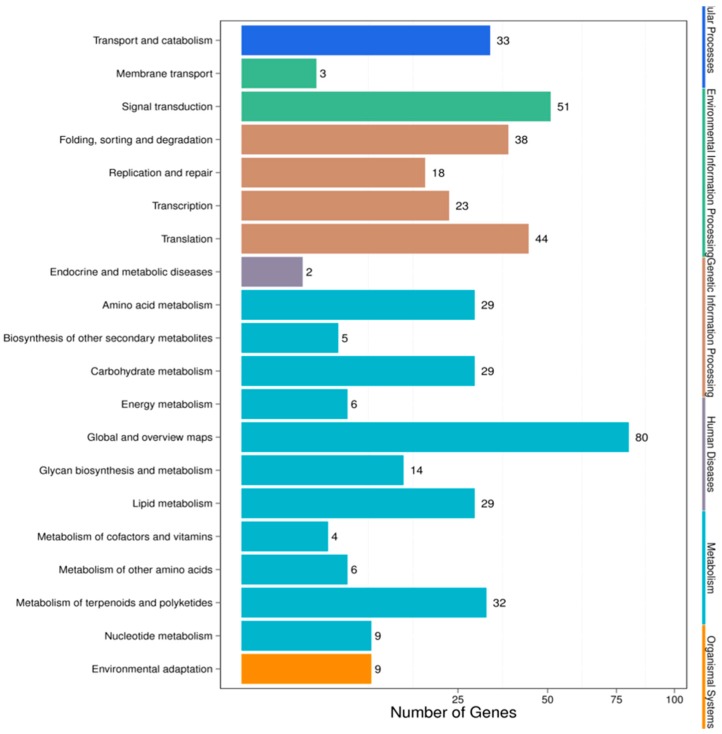
KEGG pathway classifications for the IC-1 vs. IS-4 combination.

**Table 1 genes-10-00303-t001:** Summary of the results of high-throughput sequencing of the sRNAs from the leaves of two *V. faba* genotypes growing under salt stress conditions and the control. The alignment statistics of the tags following alignment to the reference genome are summarized herein.

Sample Name	Sequence Type	Raw Tag Count	Clean Tag Count	Percentage (%)	Number of Mapped Tags	Percentage (%)
HC-4	SE50	51420111	47942551	93.24	36387118	75.9
HSA	SE50	43734868	40682196	93.02	26272525	64.58
IC-1	SE50	48229628	43865674	90.95	21451106	48.9
IS-4	SE50	53763194	41563472	77.31	20568122	49.49

**Table 2 genes-10-00303-t002:** Abundance of the reads of the different sRNA categories from the leaf samples of the two faba bean genotypes growing under control and salinity stress conditions.

Type	HC-4		HSA		IC-1		IS-1	
	Count	(%)	Count	(%)	Count	(%)	Count	(%)
Total	47942551	100	40682196	100	43865674	100	41563472	100
Intergenic	9985569	20.83	10292988	25.3	9475423	21.6	8197558	19.72
Mature (miRNA)	3224519	6.73	1653404	4.06	6841472	15.6	5633410	13.55
Rfam other sncRNA	73325	0.15	94167	0.23	24135	0.06	35172	0.08
snRNA	1405	0	4831	0.01	2381	0.01	959	0
unmap	11040957	23.03	13725210	33.74	21005681	47.89	20270917	48.77
rRNA	1773606	3.7	1100198	2.7	244021	0.56	358242	0.86
Hairpin	38	0	9	0	80	0	147	0
snoRNA	30566	0.06	14138	0.03	9971	0.02	6058	0.01
Precursor	27019	0.06	11116	0.03	24531	0.06	40935	0.1
Repeat	21771765	45.41	13683732	33.64	6237390	14.22	7018371	16.89
tRNA	13782	0.03	102403	0.25	589	0	1703	0

**Table 3 genes-10-00303-t003:** Summary of the small non-coding RNAs detected from each sample.

Sample	Known miRNA Count	Novel miRNA Count	Known piRNA Count	Novel piRNA Count	Known siRNA Count	Novel siRNA Count
IS-4	1301	51	0	0	0	2578
HSA	1118	38	0	0	0	1879
HC-4	1018	44	0	0	0	1919
IC-1	1465	48	0	0	0	2241

**Table 4 genes-10-00303-t004:** Differentially expressed conserved miRNA families.

Conserved miRNA Family	DESs in HC-4-vs.-HSA (Sensitive Genotype)		DESs in IC-1-vs.-IS-4 (Tolerant Genotype)		Target Gene Family
	Number of Upregulated Members	Number of Downregulated Members	Number of Upregulated Members	Number of Downregulated Members	
miRNA156	20	6	11	7	Squamosa promoter-binding proteins
miRNA157	2	1	2	1	Squamosa promoter-binding proteins
miRNA159	27	8	8	28	MYB transcription factors/TCP transcription factors
miRNA160	4	4	11	43	Auxin Response factors
miRNA162	9	3	8	8	Dicer Like protein/E3 ubiquitin-protein ligase RNF144A-like isoform X1
miRNA164	3	2	12	33	NAC domain protein/NAC transcription factor-like protein
miRNA165	0	1	2	2	Homeo domain-Zip transcription factors/homeobox-leucine zipper protein ATHB-14-like/bZIP transcription factor
miRNA166	39	91	82	80	Homeobox-leucine zipper protein ATHB-15-like isoform X2/bZIP transcription factor
miRNA167	38	26	27	31	Transmembrane protein, putative/translation initiation factor eIF-2B delta subunit
miRNA168	19	13	24	9	Argonautes/protamine P1 family protein
miRNA169	3	1	2	4	HAP2/NFY transcription factors/CCAAT-binding transcription factor
miRNA171	19	14	12	42	Scarecrow-like transcription factors/GRAS family transcription regulator
miRNA172	1	0	0	1	AP2 domain transcription factors/AP2-like ethylene-responsive transcription factor/myb-like transcription factor family protein
miRNA319	6	3	6	12	MYB transcription factors/TCP transcription factors
miRNA390	3	8	4	5	TAS3-primary transcripts/LRR receptor-like kinase family protein
miRNA391	1	0	0	1	TAS3-primary transcripts/zinc finger CCCH domain protein
miRNA393	8	1	2	2	F-Box protein/transport inhibitor response 1 protein/Ubiquitin
miRNA394	2	0	2	2	F-Box protein/Zinc finger CCCH domain-containing protein ZFN-like
miRNA396	36	16	31	26	Growth regulating factors/F-box protein interaction domain protein /BZIP transcription factor bZIP80
miRNA397	2	2	4	4	Laccases
miRNA398	2	7	3	13	Cu/Zn superoxide dismutases (CSD)/BAG family molecular chaperone regulator-like protein
miRNA399	3	3	8	4	Phosphate transporter/ubiquitin-conjugating enzyme E2/OBP3-responsive protein
miRNA408	1	6	16	5	Plantacyanins/uclacyanin-2-like/basic blue-like protein
miRNA2111	2	5	6	2	DNA replication factor CDT1-like protein/calcineurin-like phosphoesterase, family protein
miRNA482	0	2	1	1	

DESs-Differentially Expressed sRNAs.

**Table 5 genes-10-00303-t005:** Comparative expression profiles of the novel miRNA genes in HC-4-vs.-HSA.

miRNA ID	Count (HC-4)	Count (HSA)	TPM (HC-4)	TPM (HSA)	log2 Ratio (HSA/HC-4)	Regulation Profile (Up/Down) (HSA/HC-4)	*p*-Value	FDR
novel_mir1	193	869	5.12	31.31	2.612408	Up	1.81E-153	2.31 × 10^−152^
novel_mir5	24	124	0.64	4.47	2.804131	Up	3.67E-25	2.05 × 10^−24^
novel_mir6	0	48	0.001	1.73	10.75656	Up	1.12E-18	5.58 × 10^−18^
novel_mir7	227	1120	6.02	40.35	2.744733	Up	1.78E-208	2.63 × 10^−207^
novel_mir9	0	82	0.001	2.95	11.5265	Up	2.42E-31	1.47 × 10^−30^
novel_mir10	14	894	0.37	32.21	6.44384	Up	8.00E-307	1.47 × 10^−305^
novel_mir13	33	115	0.88	4.14	2.234055	Up	2.47E-18	1.21 × 10^−17^
novel_mir16	17	597	0.45	21.51	5.578939	Up	1.75E-194	2.51 × 10^−193^
novel_mir17	491	794	13.03	28.61	1.134682	Up	2.99E-44	2.03 × 10^−43^
novel_mir19	39	81	1.03	2.92	1.503324	Up	3.24E-08	1.14 × 10^−7^
novel_mir22	11	32	0.29	1.15	1.987509	Up	2.35E-05	7.14 × 10^−5^
novel_mir29	82	180	2.18	6.49	1.57389	Up	9.97E-18	4.82 × 10^−17^
novel_mir37	492	817	13.06	29.43	1.172133	Up	6.79E-48	4.77 × 10^−47^
novel_mir38	6941	74963	184.18	2700.76	3.874177	Up	0	0
novel_mir39	164	1013	4.35	36.5	3.068809	Up	2.05E-212	3.08 × 10^−211^
novel_mir40	31	86	0.82	3.1	1.918572	Up	1.06E-11	4.35 × 10^−11^
novel_mir41	114	2010	3.02	72.42	4.583768	Up	0	0
novel_mir42	259	1808	6.87	65.14	3.245162	Up	0	0
novel_mir43	50	130	1.33	4.68	1.815082	Up	6.45E-16	3 × 10^−15^
novel_mir45	736	2609	19.53	94	2.266969	Up	0	0
novel_mir46	143	220	3.79	7.93	1.065123	Up	3.38E-12	1.41 × 10^−11^
novel_mir48	919	1754	24.39	63.19	1.373407	Up	9.10E-129	1.04 × 10^−127^
novel_mir8	50	0	1.33	0.001	−10.3772	Down	1.20E-12	5.13 × 10^−12^
novel_mir11	926	75	24.57	2.7	−3.18587	Down	4.68E-136	5.50 × 10^−135^
novel_mir15	35	0	0.93	0.001	−9.86109	Down	4.71E-09	1.73 × 10^−8^
novel_mir20	35	0	0.93	0.001	−9.86109	Down	4.71E-09	1.74 × 10^−8^
novel_mir23	794	252	21.07	9.08	−1.21443	Down	3.39E-35	2.17 × 10^−34^
novel_mir25	59	0	1.57	0.001	−10.6165	Down	8.33E-15	3.76 × 10^−14^
novel_mir27	2970	65	78.81	2.34	−5.0738	Down	0	0
novel_mir28	16	0	0.42	0.001	−8.71425	Down	0.000168	0.000477
novel_mir33	709	173	18.81	6.23	−1.5942	Down	1.26E-46	8.65 × 10^−46^
novel_mir34	479	22	12.71	0.79	−4.00797	Down	1.78E-85	1.64 × 10^−84^
novel_mir36	41	0	1.09	0.001	−10.0901	Down	1.72E-10	6.84 × 10^−10^
novel_mir44	34	0	0.9	0.001	−9.81378	Down	8.17E-09	2.97 × 10^−8^
novel_mir53	230	47	6.1	1.69	−1.85179	Down	1.71E-19	8.64 × 10^−19^

TPM: Transcripts per Kilobase Million.

**Table 6 genes-10-00303-t006:** Comparative expression profiles of novel miRNA genes in IC-1-vs.-IS-4.

miRNA ID	Count (IC-1)	Count (IS-4)	TPM (IC-1)	TPM (IS-4)	log2 Ratio (IS-4/IC-1)	Regulation Profile (up/down) (IC-1 vs IS-4)	*p*-Value	FDR
novel_mir6	44	256	1.87	11.62	2.6354999	Up	7.82× 10^−41^	4.42 × 10^−40^
novel_mir8	338	6850	14.38	310.9	4.434315	Up	0	0
novel_mir9	0	207	0.001	9.4	13.198445	Up	5.11 × 10^−66^	3.49 × 10^−65^
novel_mir18	0	188	0.001	8.53	13.05833	Up	5.02 × 10^−60^	3.27 × 10^−59^
novel_mir19	200	1538	8.51	69.81	3.0362027	Up	3.94 × 10^−275^	5.07 × 10^−274^
novel_mir22	0	587	0.001	26.64	14.701306	Up	7.53 × 10^−186^	7.83 × 10^−185^
novel_mir30	0	200	0.001	9.08	13.148477	Up	8.25 × 10^−64^	5.54 × 10^−63^
novel_mir36	163	496	6.93	22.51	1.6996388	Up	5.04 × 10^−45^	2.99 × 10^−44^
novel_mir39	134	453	5.7	20.56	1.8508064	Up	2.78 × 10^−46^	1.67 × 10^−45^
novel_mir42	192	1707	8.17	77.48	3.245416	Up	0	0
novel_mir5	347	663	14.76	30.09	1.0275914	Up	2.31 × 10^−28^	1.10 × 10^−27^
novel_mir13	1404	298	59.72	13.53	−2.1420523	Down	2.13 × 10^−156^	1.98 × 10^−155^
novel_mir14	62	10	2.64	0.45	−2.552541	Down	8.44 × 10^−10^	2.72 × 10^−9^
novel_mir16	116	42	4.93	1.91	−1.368015	Down	2.42 × 10^−8^	7.48 × 10^−8^
novel_mir20	254	60	10.8	2.72	−1.9893528	Down	4.41 × 10^−27^	2.07 × 10^−26^
novel_mir27	10812	1387	459.9	62.95	−2.8690419	Down	0	0
novel_mir29	1387	564	59	25.6	−1.2045711	Down	1.14 × 10^−68^	7.94 × 10^−68^
novel_mir31	245	75	10.42	3.4	−1.6157486	Down	4.83 × 10^−20^	1.98 × 10^−19^
novel_mir32	4098	1680	174.31	76.25	−1.1928461	Down	3.81 × 10^−196^	4.01 × 10^−195^
novel_mir33	1370	157	58.27	7.13	−3.0307793	Down	2.35 × 10^−225^	2.65 × 10^−224^
novel_mir4	165	11	7.02	0.5	−3.811471	Down	1.52 × 10^−34^	7.93 × 10^−34^
novel_mir40	451	39	19.18	1.77	−3.4377815	Down	1.63 × 10^−84^	1.22 × 10^−83^
novel_mir44	240	0	10.21	0.001	−13.317695	Down	1.24 × 10^−69^	8.68 × 10^−69^
novel_mir46	759	263	32.28	11.94	−1.4348377	Down	1.01 × 10^−49^	6.25 × 10^−49^
novel_mir48	6461	707	274.83	32.09	−3.0983438	Down	0	0
novel_mir50	5351	1112	227.61	50.47	−2.173066	Down	0	0
novel_mir52	450	116	19.14	5.26	−1.8634561	Down	5.42 × 10^−43^	3.15 × 10^−42^
novel_mir53	871	180	37.05	8.17	−2.1810656	Down	6.75 × 10^−100^	5.38 × 10^−99^

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
