# Peer review of "Identification and Characterization of Salt-Responsive MicroRNAs in Vicia faba by High-Throughput Sequencing"

_genes, 2019, doi:10.3390/genes10040303_

Reviewer 1 Report

Although the topic itself is interesting, the method cannot be said to be described clearly. For example, for DES identification, authors denoted their methods in detail at "2.8. Screening the Differentially Expressed sRNAs (DESs)". On the other hand, there is a description "In this study, we employed the DESeq2 and ExpDiff methods for screening the DESs." at L241.  Which is correct? 

Also for target prediction "In this study, numerous software  were used to identify the possible targets" (L136) "numerous software" cannot help the readers to reproduce the results. Please describe all methods detailed enough to be able to be reproduced, although I am not wiling to list all of them.

Author Response

Reviewers-1 comments

Although the topic   itself is interesting, the method cannot be said to be described clearly. For   example, for DES identification, authors denoted their methods in detail at   "2.8. Screening the Differentially Expressed sRNAs (DESs)". On the   other hand, there is a description "In this study, we employed the   DESeq2 and ExpDiff methods for screening the DESs." at L241.  Which   is correct?

We corrected this. DESeq2 deleted from the result section

Also for target   prediction "In this study, numerous software  were used to identify   the possible targets" (L136) "numerous software" cannot help   the readers to reproduce the results. Please describe all methods detailed   enough to be able to be reproduced, although I am not wiling to list all of   them.

We corrected this paragraph and deleted “numerous software  were used to identify   the possible targets" (L136)”

Reviewer 2 Report

The authors performed a genome-wide survey of miRNAs possibly involved in the salt response of Vicia faba by RNA-seq. Although the topic itself is of significance, the manuscript is not very informative. The experimental design and data presentation need to be substantially improved. 

My major concerns include:

 The current manuscript lacks experimental validation of the differentially expressed miRNA and its putative targets. At least, the most important salt-responsive miRNA families and their corresponding targets identified in this study should be validated by qRT-PCR.

Functional analysis (GO and KEGG) of miRNA putative targets is too general, and didn't provide details regarding which physiological/cellular/metabolic processes are, in particular, responsive to the salt treatment.

The authors could further improve the manuscript by comparing the salt-responsive miRNA profiles between the salt-sensitive and salt-tolerant genotypes. This will help to identify the particular miRNA families that contribute to the different salt responses of the two genotypes. 

Here are some detailed comments:

Introduction:

Literature is not up-to-date. Recent research progresses in the miRNA-regulated salt responses should be included.

Materials and methods:

How frequent was the salt solution applied to the plants? 

Does the salt treatment of  35 days have literature support? I will assume this treatment is too long that the secondary responses/global responses were already induced, resulting in some of the miRNA detected in the study was not directly involved in salt stress, but other secondary responses.

Were all the existing leaves on the plants collected for RNA extraction, or instead, particular leaves were collected?

Result:

Change the sample names to something more indicative like"salt-sensitive-control, salt-sensitive-salt, salt-tolerant-control, salt-tolerant-salt" will make it easier to understand.

Why the percentages of mapped tags in IC-1 and IS-4 samples (~50%) are much less than those of HC-4 and HSA (~70%)? is this due to the genotype differences or sRNA library construction issue?

Discussion:

How was the salt stress-related miRNAs identified in this study compared to those in other plant species, particularly in legume plants? This will provide more insights into the conservations of salt stress-related miRNAs in the phylogenetic context.

Author Response

Reviewers 2 comments

Rebuttal

 The current manuscript lacks experimental validation of   the differentially expressed miRNA and its putative targets. At least, the   most important salt-responsive miRNA families and their corresponding targets   identified in this study should be validated by qRT-PCR.

As this is a first report of miRNA in faba bean, so we are   reporting only initial findings of this study.

We will also do the validation of important salt   responsive miRNA families   and their corresponding targets in details by qRT-PCR   and would like to report these in separate article.

Functional analysis   (GO and KEGG) of miRNA putative targets is too general, and didn't provide   details regarding which physiological/cellular/metabolic processes are, in   particular, responsive to the salt treatment

Details have been given in Supplementary Table S6.

The authors could   further improve the manuscript by comparing the salt-responsive miRNA   profiles between the salt-sensitive and salt-tolerant genotypes.   

The numbers of differentially expressed miRNA families have been described   in Table 4 by   comparing the salt-responsive miRNA profiles between the salt-sensitive and   salt-tolerant genotypes.    

Literature is not   up-to-date. Recent research progresses in the miRNA-regulated salt responses   should be included

It was updated. Latest references were added

How frequent was the salt solution applied to the plants?

Does the salt treatment of  35   days have literature support? I will assume this treatment   is too long that the secondary responses/global responses were   already induced, resulting in some of the miRNA detected in the study   was not directly involved in salt stress, but other secondary responses.

Samples were taken after two weeks of salt treatments. It was   incorporated.

Were all the existing leaves on the plants collected for RNA   extraction, or instead, particular leaves were collected?

All leaves were extracted and grounded for RNA isolation.  We 100 mg powder for RNA extraction

Change the sample names to something more indicative   like"salt-sensitive-control, salt-sensitive-salt,   salt-tolerant-control, salt-tolerant-salt" will make it easier to   understand.

Generally the given names were used.

Why the percentages of mapped tags in IC-1 and IS-4 samples (~50%) are much less   than those of HC-4 and HSA (~70%)? is this due to the genotype differences or   sRNA library construction issue?

It might be due to genotype differences.

How was the salt stress-related miRNAs identified in this study   compared to those in other plant species, particularly in legume plants? This   will provide more insights into the conservations of salt stress-related   miRNAs in the phylogenetic context

One comparative paragraph was added

Round  2

Reviewer 1 Report

The authors addressed all the concerns.

Reviewer 2 Report

Thanks to the efforts the authors have made to improve the manuscript. Most of my concerns were addressed in this version. Although the authors didn't supply experimental validation of the differentially expressed miRNA and its putative targets, it is still acceptable considering the adequate information this study provided. I think the current version is ready for publication on Genes.